# Applicability of Fuzzy Logic and Artificial Neural Network for Unpaved Airfield Surface Bearing Strength Prediction

**DOI:** 10.3390/s21103373

**Published:** 2021-05-12

**Authors:** Ludek Cicmanec

**Affiliations:** Department of Air Force, Faculty of Military Technology, University of Defence in Brno, 66210 Brno, Czech Republic; ludek.cicmanec@unob.cz; Tel.: +420-604-469-251

**Keywords:** airport, airfield, surface, unpaved, bearing strength, prediction model, fuzzy logic, artificial neural network

## Abstract

The main objective of this paper is to describe a building process of a model predicting the soil strength at unpaved airport surfaces (unpaved runways, safety areas in runway proximity, runway strips, and runway end safety areas). The reason for building this model is to partially substitute frequent and meticulous inspections of an airport movement area comprising the bearing strength evaluation and provide an efficient tool to organize surface maintenance. Since the process of building such a model is complex for a physical model, it is anticipated that it might be addressed by a statistical model instead. Therefore, fuzzy logic (FL) and artificial neural network (ANN) capabilities are investigated and compared with linear regression function (LRF). Large data sets comprising the bearing strength and meteorological characteristics are applied to train the likely model variations to be subsequently compared with the application of standard statistical quantitative parameters. All the models prove that the inclusion of antecedent soil strength as an additional model input has an immense impact on the increase in model accuracy. Although the M7 model out of the ANN group displays the best performance, the M3 model is considered for practical implications being less complicated and having fewer inputs. In general, both the ANN and FL models outperform the LRF models well in all the categories. The FL models perform almost equally as well as the ANN but with slightly decreased accuracy.

## 1. Introduction

Aviation may be one of the most constrained spheres of human activities that are pushed to their limits. Engineers are forever improving technologies covering aircraft designs and material properties. It is expected that not only an extended range of flight, greater load capacity, and reduced fuel consumption but also the safety standards are pushed to new levels. The technologies applied at airports are not the exception. As some airport constructions have reached into the adoption of building information modeling (BIM) [1], the digitalization and integration of the airport movement area design might be considered as well. Basically, the airport design has an immense effect on incidents and accidents occurring at airports. While investigating the aviation safety databases, it was figured out that a large proportion of aircraft accidents occur at airports and particularly nearby runways. The majority of them are associated with runway excursions, accidental aircraft run-off, and over- and under-shooting a runway [2]. Therefore, as to address this, Ranieri et al. proposes a gradual bearing strength decrease (17, 16, and 15 CBR) in three areas along a runway, 30–50, 50–65, and 65–105 m distant from the runway centerline [3]. The aviation authorities regularly amend safety standards focusing on the design of the close vicinity of runways. Apart from obstacle-free zones, they issue standards in the quality of unpaved areas adjacent to runways [4,5]. The bearing strength is supposed to be evaluated regularly, being exceedingly dependent on actual meteorological conditions. The daily bearing strength measurements are expected to be maintained over the entire airport vicinity to retain an airport operational [6]. Due to airport operation restrictions imposed mainly by extensive time constraints in daily operations, it is vital to conduct the bearing strength evaluations as a time reference at two locations at a minimum every day [6]. Despite being seemingly highly time-efficient, these measurements are exceedingly obstructive and time-consuming for airport operation staff. As a consequence, a kind of prediction model giving, to some extent, prospect of the unpaved surface condition seems to be essential. This should not entirely substitute the daily measurements because these values should be officially reported. It should work as an aid for airport operation staff to plan and organize the surface maintenance.

Firstly, soil strength is remarkably vital in geotechnical engineering and building practice. It is one of the paramount properties which highway and airport foundations are designed to. A great deal of research targeting mechanical properties, consolidation, and strength has been conducted. The investigation of soil behavior and its application as construction materials was launched by Whitlow [7]. It was figured out that the problem is rather complex due to a fair amount of diversity in physical properties in soils [8]. Currently, as a common practice, soil strength tests, namely the California bearing ratio (CBR) tests are conducted in a laboratory or in situ to target field applications [9,10]. However, credible as these tests are, they are rather prolonged, elaborate, and costly. Despite being tested at airports, as shown by Detsky et al. [11], the applicability is feasible for the surface layer only. This was proven to be unacceptable for daily airport operations since the strength profile is needed for a deeper soil layer [12]. As a consequence, penetrometer applications, which can freely assess the soil profile down to 1 m, have been utilized not only in building practice but also in airport operations to report the actual bearing capacity of unpaved surfaces. The Dynamic Cone Penetrometer (DCP), having been altered for airfield applications, is an example. The field test incorporating the DCP is accomplished to ensure the required property of various surfaces consisting of unpaved, unprepared runways, runway strips, and runway safety areas (RESAs). Being fairly time- and labor-intensive, the daily measurements might be supplemented by a soil strength prediction model. Such a model should provide essential information over the particular surface condition, serving to organize surface maintenance.

A great deal of studies have been dedicated to soil conditions, specifically to the bearing strength. From the outset, it was known that development and changes in bearing strength in time are highly non-linear, time-varying, spatially distributed, and therefore not easily formalized. Both analytical and empirical approaches have been tested with more or less success.

The analytical approach is knowledge-based. Generally, it focuses on describing systems with physical laws. That is to say, it uses a mathematical framework based on mass, momentum, and energy conservation spatially distributed equations to formalize soil parameters associated with the strength. An example of such a model was implicated in predicting the cohesion and friction angle at the penetrometer cone [12]. Then, a soil moisture prediction strength model (SMSP) has been imposed and refined several times by implying coefficients to predict the cone index (CI) and rating cone index (RCI). It applies real-time meteorological data to up-to-date forecasts. Regarding the model accuracy, the relations were considered non-smooth, which may cause some disparity in the estimation of the required physical parameters [13,14,15].

As the physical models have difficulties formalizing complex and variable systems, an empirical approach (clearly statistical) is essential to be implicated. The prediction of wave velocities for the cone penetration test with the linear regression is currently considered state-of-the-art [16]. Following this, a relationship between the RCI, moisture, and clay content was well established by a statistical approach and machine learning [17]. More recently, a relation between the terrain and the vehicle track was ascertained by Baladi and Rohani [18]. On top of this, NATO utilizes its NATO reference mobility model (NRMM) that predicts the bearing capacity based upon rainfall and terrain. The data set comprising location, precipitation, and evaporation is matched against traction and the vehicle cone index (VCI) [19].

Thirdly, a soil strength prediction model might be based on soil moisture forecast, as these models are broadly available with considerable accuracy [20]. The only obstacle is the moisture–strength correlation. Regarding practicability and high reliability, plenty of observations in a full soil moisture range are needed [21].

Fourthly, machine learning should also be considered, as it has been successfully implemented in many spheres of human activities. For instance, neural network and fuzzy logic (FL) models predict material mechanical properties in engineering [22,23]. On the other hand, in geotechnics, these models predict soil properties based upon meteorological characteristics. The surface temperature is given by the application of brightness [24,25] or low-level atmospheric parameters such as relative humidity and temperature to predict soil moisture using an artificial neural network (ANN) [26]. Particularly here, backpropagation learning was successfully tested [26]. It is also worth considering the support vector machines since they displayed better performance than the ANN while comparing the performance forecasting soil moisture [27]. Having directly sought bearing strength models, the ANN and FL were used to predict a peak shear angle of soil [28], the bearing strength, and behavior of shallow foundations [29]. Rather interesting is the idea of predicting the soil strength directly from soil properties, such as water content, plasticity index, dry density, and soil composition [30]. Specific soft soils were tested in an ANN model predicting a compression coefficient [31].

The hydrology models are rather significant in rainfall and flood prediction. Generally, they can be characterized by a great deal of complexity and time variability. Many successful implications were mounted with the ANN and FL support. The majority of these applications proved to be highly practical, facilitating better accuracy in comparison to traditional models such as linear regression models [32,33,34,35].

In short, while seeking the optimal solution for bearing strength prediction, the problem is complex and has too much time variance for a physical model. However, the statistical models, represented primarily by ANN and FL models, were anticipated to provide sufficient accuracy. In order to tackle the soil prediction at airfields, ANN and FL approaches were supposed to be mounted and tested upon a linear regression function (LRF). The results should work as a base for an intelligent model predicting the unpaved airport surface bearing capacity.

## 2. Materials and Methods

In general, the idea was to set up sets of models with varying input data based on LRF, an ANN, and FL to predict the bearing strength of both unpaved runways and unpaved surfaces adjacent to paved runways. The performance was considered to evaluate with validation indicators such as a correlation coefficient (R), mean square error (MSE), and root mean square error (RMSE).

### 2.1. Subjects

Experimental data were determined from a single airport for the past four years (2016–2020). That is to say, 1400 sample sets of both meteorological and soil strength values were selected for the study. The data sample is reasonably representative as it covers periods with extremes, including intense heat and excessive precipitation. This particular airport was assigned due to the high intensity of traffic with varying aircraft weight categories. The traffic is executed on both the paved and unpaved runways with dimensions 2500 times 45 m and 1000 times 40 m. As the traffic involving heavy aircraft is conducted over the unpaved runway, the airport operation requires frequent inspections comprising bearing strength measurements and monitoring. On the other hand, concerning the paved areas, runways, and taxiways, a likelihood of accidental excursions always exists. Therefore, the bearing strength is measured and monitored at predefined reference locations along the paved runways.

The airport geographical location with elevation varying from 450 to 475 m above sea level is rated as low or medium altitude locations in the Bohemia-Moravia Highland. The soil profile in the vicinity of the runway is characterized by white gneiss bedrock, whereas the surface, which is approximately 2 m thick, is covered by eroded gneiss. Closer to the surface, it turns into silty sands and into silts in some separate locations. At the reference points at the airfield, the soil profile consists of the top 40 cm layer with well-graded silty sand and a 60 cm sublayer with well-graded gravel with sand. With regard to hydrological conditions, the place is weak in underground water. Being notably eroded, the silty surface does not create sufficient conditions for larger water amount catchments.

While seeking the most suitable inputs into the statistical model, the meteorological data were considered. Predominantly, attention was dedicated to a daily precipitation rate and daily mean air temperature. The surface synoptic observation (SYNOP) data were chosen as the most suitable and reliable source. The records being recorded by either automated weather stations or manually every six hours incorporate a numerical code for regular observation reporting. They describe general weather information comprising temperature, barometric pressure, and visibility at the station.

Coming to the model output, as mentioned, it should represent the unpaved surface properties, particularly the bearing capacity. As the operational requirement for a field application at the particular airport is to report the strength value in MPa, the traditional methodology for bearing strength evaluation applied at Czech military airfields was maintained. The daily operation measurements conducted regularly every morning at two reference points were considered appropriate for the study. A device that is required by this methodology is the aerodrome dynamic penetrometer (ADP) (see Figure 1). As a matter of fact, the actual measured values are estimates of the surface bearing capacity in pressure units (MPa). The pressure is a measure of aircraft undercarriage characteristics and weight, which is correlated with a number of blows into specific soil depths [36]. Having already discussed the aviation authority requirements, which generally accept the California bearing ratio (CBR) for reporting in aviation applications, the device is correlated with the CBR, issuing CBR estimates [6].

In short, a specific airport in the vicinity of the Czech Republic was chosen for the study. Specifically, daily measurements of the bearing strength utilizing the ADP were decided, whereas the basic meteorological data were considered to be obtained from the SYNOP reports. The four-year data set collection was then divided into two. The first was dedicated to model calibration whilst the second to validation.

### 2.2. Procedure

Firstly, considering the model design concept, a selection of input and output variables is the first step in model development. In this particular case, the bearing strength is the function of meteorological, topographical, soil, and vegetation conditions. Theoretically, a non-linear, time-varying function may be suitable to express the soil conditions. Nevertheless, insufficient data to execute this were available. The whole prediction relied on the rainfall data and mean air temperature. For input consideration, the potential candidate models were expected to be constructed and compared with quantitative statistical tools.

Based on expert analysis, the models with the following inputs were considered for further examination:
Precipitation rate and mean air temperature.
(1)$$\;\sigma_{t}=f\left(P_{t},T_{t}\right),$$
(2)$${\;\sigma }_{t}=f\left(P_{t},P_{t7},T_{t},T_{t7}\right),$$Precipitation rate and antecedent soil strength.
(3)$$\sigma_{t}=f\left(P_{t},\sigma_{t-1}\right),$$
(4)$${\;\sigma }_{t}=f\left(P_{t},{\;P}_{t-1},\sigma_{t-1}\right),$$
(5)$$\sigma_{t}=f\left(P_{t},{\;P}_{t7},\sigma_{t-1}\right),$$Precipitation rate, mean air temperature, and antecedent soil strength.
(6)$$\sigma_{t}=f\left(P_{t},{\;T}_{t},\sigma_{t-1}\right),$$
(7)$${\;\sigma }_{t}=f\left(P_{t},P_{t-1},T_{t},\sigma_{t-1}\right),$$
(8)$${\;\sigma }_{t}=f\left(P_{t},P_{t-1},T_{t},T_{t-1},\sigma_{t-1}\right),$$
(9)$${\;\sigma }_{t}=f\left(P_{t},P_{t7},T_{t},T_{t7},\sigma_{t-1}\right),$$
(10)$${\;\sigma }_{t}=f\left(P_{t},P_{t7},T_{t},T_{t7},\sigma_{t-1},\sigma_{t-2}\right),$$
where P_t_ refers to a daily precipitation rate (P_t−1_ to a daily precipitation rate minus one day), T_t_ depicts daily mean air temperature (T_t−1_ is mean air temperature minus one day) whereas σ_t−1_ is antecedent soil strength minus one day (σ_t−2_ is antecedent soil strength minus two days).

Out of the potential meteorological information, a daily precipitation rate (P_t_) and daily mean air temperature (T_t_) were selected as the primary inputs which should address the prospect of future conditions. On top of these two, an additional one representing the actual surface state and the initial conditions was considered. During the initial thoughts, the soil moisture content was anticipated as the antecedent soil condition. However, by the time at least half of the experiment had been executed, the measured soil moisture was realized to be applicable for a limited surface layer. As a consequence, the soil moisture data set was figured out as insufficient. Instead, the soil strength representing the preceding conditions (antecedent soil strength σ_t−1_) was considered. The soil moisture issue and the idea of refining the entire model are reiterated in the discussion part of the paper.

To sum up, as mentioned in the previous text, the first important step in building any model is input variables selection. Aiming at the soil strength prediction, the soil strength is considered a function of previous rainfall, run-off, meteorological, topographical, and soil conditions. Theoretically, a water storage function could possibly be applied to express water flow through the location. However, no sufficient data are present in doing so, but for rainfall data. Therefore, a combination of basic meteorological characteristics and the introduction of antecedent soil strength were selected to create input to build the bearing strength prediction model.

### 2.3. Linear Regression Function

Generally, in statistics, the main objectives are to build statistical models that resemble real objects in some respect. Such a process is carried out through regression analysis that explains likely relations between variables. LRF can be expressed as [37,38]:(11)$${\;y}_{i}=\beta_{0}+\overset{k}{\underset{k=1}{\sum }}\beta_{k}\times f_{k}\left(x_{i1},x_{i2},\ldots ,x_{\mathrm{ip}}\right)+\varepsilon_{i},$$
where y_i_ represents i-th response or prediction, whereas xi i-th observation. Then, b_k_, ε_i_ and f_k_ depict k-th coefficient, random error, and scalar-valued function of the x_ij_ predictor variable.

### 2.4. Fuzzy Logic Model

Since FL has become increasingly popular in commercial applications ranging from washing machines, cameras, industrial control, medical controls, etc., the second base model concept targeting soil strength predictions was dedicated to fuzzy logic.

The basic structure of fuzzy interference systems is a model that copies input variables to membership functions, and copies these into rules which model an output value throughout output characteristics and membership functions. The membership functions are chosen randomly, whereas the rules are user interpretations. Since it possesses a broad series of input and output data, the model structure does not necessarily have to be predetermined. Therefore, these membership functions may change throughout the learning process. Backpropagation or a combination of least squares estimation and backpropagation for membership function estimation is applicable. That is to say, this modeling, providing fully representative data, is suitable to present the expected model. However, in practice, noisy data should be anticipated as well. On account of mitigating their effects, usually, the validation is implemented. The whole validation idea is to expose the model to additional input/output data, which the model is not trained to. The outcomes subsequently quantify how well the model performs [39,40,41,42].

While creating the bearing strength prediction model based on FL, the adaptive neuro-fuzzy interference system (ANFIS) was applied to train a Sugeno-type fuzzy model in the MATLAB environment. The hybrid learning algorithm uses a combination of least-squares and backpropagation for training. A checking data set helps to validate the model for overfilling the training data [39]. This approach has gained a fair amount of popularity due to uncomplicated construction, which can deal remarkably well with imprecision and uncertainty. FL classifies objects with vague input and output variable margins. These are linked with the aid of membership functions to create a logical system [39].

### 2.5. Artificial Neural Network Model

The third approach is dedicated to the ANN, since it has earned the most popularity in control of rather complex problems such as pattern recognition, system identification and classification, speech and vision recognition, and in general, it is efficient in function fitting. The whole idea anticipates a system inspired by a nervous system that is composed of elements and connections. The connections, sometimes referred to as weights, determine the network functions. While controlling the large data sets to cover all inputs and outputs, the weight can be adjusted by specific training, leading the inputs well to the precise target outputs. As a result, a reasonably simplistic neural network can fit any practical function. By and large, the network is separated into layers. The weights are imposed only between elements in different layers. A standard network is a two-layer feed-forward neural network with hidden neurons (transfer function) and linear output neurons [43,44]. In all the algorithms, usually, 70% of the data are dedicated to training. During this time, the network is adjusted to its error. The rest is assigned to validation and testing. The validation is applied to measure neural network generalization, and it will stop the training when generalization stops improving. The testing does not affect the training. The entire performance may be enhanced by an increase in the number of hidden neurons. Nevertheless, the recommended 10 hidden neurons were sufficient to acquire suitable training for all input/output combinations.

To summarize, as shown in the previous text, the availability of a comprehensive data set of the bearing strength measurements of unpaved airport surfaces enables an assembly of a statistical model. A total 10 input combinations based upon meteorological characteristics, predominantly by a precipitation rate and daily mean air temperature, were suggested. LRF, FL, and ANN models were decided to take into consideration. The conception of all three model types was briefly described.

### 2.6. Statistical Analysis

A comparison of all the designated models was based on performance evaluation through the quantification of three efficiency terms containing a mean square error (MSE), correlation coefficient (R), and root mean square error (RMSE). The MSE measures the average of the squares of the errors. It is considered as a measure of the quality of an estimator. It is always non-negative, and with regard to the rating, the values closer to zero are better [45]. The R is a measure of the strength and direction of the linear relationship between two variables. It tests the goodness of fit by executing a linear regression between predicted and measured values [46]. Finally, the RMSE was applied to match the difference with reality. It represents the quadratic mean between predicted and observed values [47]. The MSE, R, RMSE can be obtained using Equations (12)–(14). All were calculated by a custom script written in the programming environment.
(12)$$\mathrm{MSE}=\frac{1}{n}\overset{n}{\underset{i=1}{\sum }}{\left(X_{i}-Y_{i}\right)}^{2},\;$$
(13)$$\;R=\frac{\sum_{i=1}^{n}X_{i}Y_{i}}{\sqrt{\sum_{i=1}^{n}X_{i}^{2}\sqrt{\sum_{i=1}^{n}Y_{i}^{2}}}},$$
(14)$$\mathrm{RMSE}=\sqrt{\frac{1}{n}\overset{n}{\underset{i=1}{\sum }}{\left(X_{i}-Y_{i}\right)}^{2}},$$
where X_i_ represents a vector of observed values of the variable being predicted, whereas Y_i_ depicts a vector of predicted values.

## 3. Results

Models for the unpaved airport surface bearing strength prediction were developed, utilizing LRF, FL, and ANN models with different input data vector combinations. The statistical evaluation criteria such as the MSE, R, and RMSE were compared to test the model performance.

### 3.1. Linear Regression Function Model

The LRF models were developed for the prediction of soil strength. The LRF models were designed with the same input and output data set as the FL and ANN models to support the relevant comparison. Concerning various input vector combinations (Equations (1)–(10)), 10 model arrangements were developed and quantified (Equations (15)–(24)).
Precipitation rate and mean air temperature.
(15)$$\sigma_{t}=0.896-0.024\times P_{t}+0.016\times {\;T}_{t},$$
(16)$$\sigma_{t}=0.893-6.96\times 10^{-5}\times P_{t}-0.001\times P_{t7}+0.11\times T_{t}+0.0018\times T_{t7},$$Precipitation rate and antecedent soil strength.
(17)$$\sigma_{t}=0.098-0.005\times P_{t}+0.919\times \sigma_{t-1},$$
(18)$$\sigma_{t}=0.1-0.005\times {\;P}_{t}-0.001\times P_{t-1}+0.918\times \sigma_{t-1},$$
(19)$$\sigma_{t}=0.089-0.006\times P_{t}+7.267\times 10^{-4}\times P_{t7}+0.921\times {\;\sigma }_{t-1},$$Precipitation rate, mean air temperature, and antecedent soil strength.
(20)$$\sigma_{t}=0.12-0.005\times P_{t}+0.002\times T_{t}+0.871\times \sigma_{t-1},$$
(21)$${\;\sigma }_{t}=0.123-0.005\times P_{t}-0.001\times {\;P}_{t-1}+0.002\times T_{t}+0.869\times \sigma_{t-1},$$
(22)$${\;\sigma }_{t}=0.125-0.005\times P_{t}-0.002\times P_{t-1}+0.001\times T_{t}+0.002\times T_{t-1}+0.866\times \sigma_{t-1},$$
(23)$${\;\sigma }_{t}=0.122-0.006\times {\;P}_{t}+3.001\times \;10^{-4}\times {\;P}_{t7}+1.067\times \;10^{-4}\times T_{t}+0.003\times T_{t7}+0.864\times {\;\sigma }_{t-1},$$
(24)$${\;\sigma }_{t}=0.106-0.006\times {\;P}_{t}+2.531\times 10^{-4}\times P_{t7}+3.964\times 10^{-4}\times T_{t}+0.002\times {\;T}_{t7}-0.729\times \sigma_{t-1}+0.155\times \sigma_{t-2},$$

The performance of the LRF presented in Equations (15)–(24) is displayed in Table 1.

Model comparisons were conducted with the aid of the MSE, R, and RMSE. The LRF models displayed the MSE in the range of 0.027 and 0.005, whereas the R was in the interval of 0.34 and 0.88. The values of the RMSE varied from 0.166 to 0.072, while having added antecedent soil strength, significant performance improvements were achieved. The M2 model showed the best performance with the MSE, R, and RMSE reaching only 0.024, 0.418, and 0.155, respectively, in the first model group, where a precipitation rate and mean air temperature were only inputs. On the other hand, the following two groups seemed to be more progressive. In the second group with a precipitation rate and antecedent soil strength as inputs, the M6 model surpassed the others as the MSE, R, RMSE reached up to 0.005, 0.88, and 0.070. Finally, the models considering a precipitation rate, mean air temperature, and antecedent soil strength as inputs displayed further slight improvement. The M10 model beat the others with the MSE, R, and RMSE as 0.005, 0.881, and 0.069, although the high resolution was achieved by the other two models, the M7 and M8 having almost identical values. Figure 2, Figure 3, Figure 4 and Figure 5 display the time series of the observed and predicted soil strength with the LRF models for the entire validation period.

To summarize, in the entire LRF arrangement, the best performance was achieved by the M8 and M9 models, which, however, do not significantly differ from the M3 model. In contrast, this particular model is characterized by its simplicity earned by fewer inputs, and therefore unnecessary complications. As a consequence, the M3 model seems to be optimal for practical application.

### 3.2. Fuzzy Logic Model

FL models were developed with the structure outlined in Equations (1)–(10). The outcomes of these specific models were compared among themselves and with the other FL and ANN models. As quantitative parameters, the MSR, R, and RMSE were applied. Table 2 displays the model performances from both calibration and validation data sets.

Firstly, the fuzzy logic bearing strength prediction models displayed the MSE ranging from 0.004 to 0.026 in the calibration and 0.006 to 0.03 in the validation period. The R was from 0.618 to 0.959 in the calibration phase, and from 0.405 to 0.943 in the validation. The RMSE ranged from 0.173 to 0.057 for the calibration and from 0.173 to 0.058 for the validation phase.

Secondly, the quantitative statistical characteristics varied slightly within each model group. In the first group with a precipitation rate and mean air temperature as the only inputs, the M2 model performed much better than the M1 since the MSE equaled 0.02 in the calibration and validation phase. The R reached 0.712 and 0.638 in the calibration and validation periods. The RMSE was 0.142 for both calibration and validation. In the second model group, where the antecedent soil strength was considered along with a precipitation rate input, the M5 model outperformed the other models, although the difference, particularly from the M3 model, was pretty insignificant. Here, the MSE was 0.004 for both calibration and validation. The R was between 0.953 and 0.934, and the RMSE reached the margin of 0.061 for both calibration and validation. In the third group with a precipitation rate, mean air temperature, and antecedent soil strength, the M6, and M7 models outnumbered the other models within the group. The MSE reached up to 0.003 for both calibration and validation. The R displayed a great deal of accuracy with 0.96 and 0.94 for the calibration and validation periods. The RMSE equaled 0.058 and 0.057.

Finally, while comparing the overall results with the other model groups, both the groups with antecedent soil strength with the MSE, R, and RMSE equaling roughly 0.04, 0.95, and 0.06, were much superior to the first group with the MSE, R, and RMSE which were about 0.022, 0.62 and 0.15. Out of the two model groups with antecedent soil strength as inputs, the models with a precipitation rate and mean air temperature displayed slight performance improvement. Figure 6, Figure 7 and Figure 8 display the time series of observed and predicted soil strength for three FL models for the entire validation period.

Out of all FL models, the M7 and M6 models gained the best outcomes. The M5 and M3 models performed almost equally well, however, with fewer inputs.

### 3.3. Artificial Neural Network Bearing Strength Prediction Model

As in the previous models, the artificial neural network models were constructed to predict unpaved airport surface bearing capacity considering Equations (1)–(10). In order to formulate performances, statistical characteristics such as the MSE, R, and RMSE were proposed for each one, using 70% of data for calibration and the remaining 30% of data for validation. The most important results are presented in Table 3 for both periods. Not only were the model comparisons conducted within the entire ANN model group, but also, they were compared with the other two principally different models.

In general, the ANN models displayed the MSE in the 0.003 and 0.026 margins. The R was from 0.456 to 0.962, and the RMSE in the range of 0.055 and 0.16.

In the models being compared within the three main groups, the M2 model outperformed the M1 model. The MSE, R, and RMSE were 0.021, 0.71, and 0.143 (0.017, 0.699, and 0.13 for validation) in the first group with a precipitation rate and mean air temperature as only inputs. In the second group concerning a precipitation rate and antecedent soil strength input combination, the M5 model displayed the best resolution. The MSE, R and RMSE were 0.003, 0.961 and 0.056 (0.005, 0.945 and 0.056 for validation). However, the difference in performance from the M3 and M6 models was minuscular. Out of the last group with a precipitation rate, mean air temperature, and antecedent soil strength inputs comprising the M7, M9, and M10 models showed nearly equivalent performances. The M8 model displayed the MSE, R, and RMSE as 0.003, 0.962, and 0.056 (0.005, 0.922 and 0.069 for validation).

Regarding comparisons in the entire ANN group, the outcomes of adding antecedent soil strength as additional model input had an enormous impact on the accuracy. The example is a development of the R reaching 0.67 in the first group without antecedent soil strength whereas 0.96 on average in the second and third groups with antecedent soil strength. Although the M7, M9, and M10 third group models displayed the best resolution, the second group performed with sufficient accuracy but fewer inputs. Figure 9, Figure 10, Figure 11 and Figure 12 show the time series of observed and predicted soil strength from the ANN models for the entire validation period.

### 3.4. General Comparison of Tested Methods

Concerning the main study objectives, which was to investigate the capacity of FL and ANN models to carry out the soil strength prediction, the models based on FL and ANN were compared with LRF. The same data sets were applied for both model calibration and validation. So as to provide appropriate comparisons, statistical tools such as the MSE, R, and RMSE were employed. The results were graphically displayed in the MSE (Figure 13a,b), R (Figure 14a,b), and RMSE charts (Figure 15a,b) for the FL, ANN, and LRF models.

In all the models, it is visible that the inclusion of antecedent soil strength as an additional model input had an immense impact on increasing model accuracy. Specifically, the best MSE, R, and RMSE were 0.02, 0.71, and 0.14 for the model without antecedent soil strength input, whereas they were 0.005, 0.96, 0.06 with. Having the models tested within groups, the M10 model displayed the best resolution for the LTF with the MSE, R, and RMSE as 0.005, 0.881, and 0.069, whereas the M7 model, whether the ANN or FL, displayed the MSE, R, and RMSE as 0.003, 0.96, and 0.056. In total, the M7 model set up with the ANN is reported as the most precise model for bearing strength prediction of soils. However, considering practicability, the M3 model is essential due to the lower input presence than in the M7 model, despite displaying slightly worse accuracy. The FL and ANN models outperformed the LRF models well in all the classes. The ANN and FL models performed similarly during training and validation, even though the ANN models displayed slight superiority over the FL models in most cases.

## 4. Discussion

A range of different models were developed to investigate the impacts of likely inputs on the model performance. The application of FL and ANN displays prospects for statistical training theory utilization in predicting complex processes. The idea has a rigid mathematical base. Therefore, it can clearly address other applications. Although the model is considered a success from a practical point of view, from a theoretical view, it can hardly be generalized since it is still a local statistical model. Looking at the current bearing strength prediction models, they do not predict the strength directly. They are based on hydrological, run-off, water catchment models predicting the soil moisture. The bearing strength is then evaluated upon soil strength and moisture correlations. As mentioned, the original idea was to mount a model with the aid of moisture measurements. However, the moisture data collection relied entirely on the soil measurements in the top layer. This was later found insufficient for a correlation with the strength measured and recorded with greater accuracy in layers down to 60 cm. As a consequence, the moisture measurement concept was due to be refined. The original measurement in a reference depth was exchanged with soil measurements in the same pattern as the soil strength is to support comprehensive information over the entire soil profile.

The future work should concentrate on a model incorporating the soil moisture as an additional input and output representing antecedent soil conditions (see Figure 16).

With the soil moisture and data being collected in layers, the prediction should address the same layers too. The current ongoing project has been working in a specific airport condition distinguished by an overall silty sand soil condition. The future project should also cover a broader range of soils. An apparent interest is supposed to be taken into soils with higher silt content. It is aiming to generalize the experience into a more complex model permitting different soil layers.

## 5. Conclusions

The availability of a detailed and immense data set containing the bearing strength measurements of unpaved airport surfaces enabled one to build a statistical model for strength prediction. The ANN and FL were selected as powerful tools to produce the series of models to be subsequently compared with the LRF models. Ten input combinations based upon meteorological characteristics (a precipitation rate and daily mean air temperature) were considered. Having assessed the model performance with the aid of statistical characteristics (MSE, R, RMSE), all the models proved that the incorporation of antecedent soil strength as an additional model input had an immense impact on the increase in model accuracy. Generally, the M7 model set up with the ANN displayed the best performance (MSE, R, and RMSE as 0.003, 0.96, and 0.056). However, the M3 model showing a slightly decreased performance was anticipated for practical implications due to fewer inputs. The ANN and FL models performed almost equally well, although the ANN was a little superior. Both the ANN and FL outnumbered the LRF models well in all categories.

## Figures and Tables

**Figure 1 sensors-21-03373-f001:**
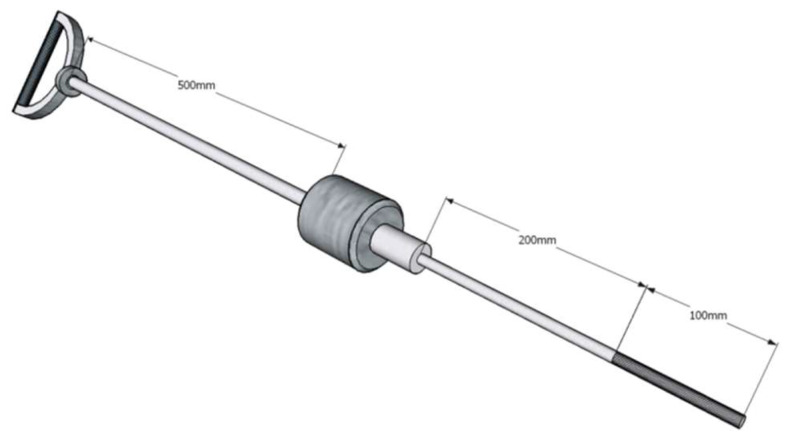
A scheme of the aerodrome dynamic penetrometer (ADP).

**Figure 2 sensors-21-03373-f002:**
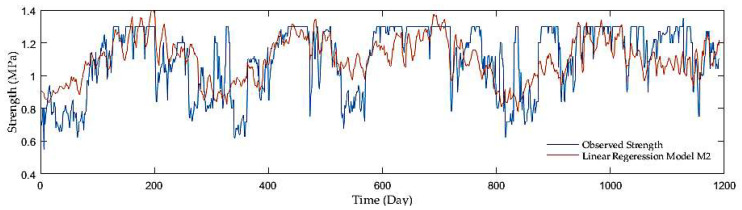
A time series of observed soil strength and predicted by the LRF model (M2).

**Figure 3 sensors-21-03373-f003:**
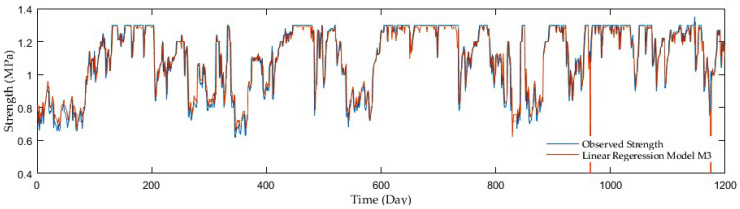
A time series of observed soil strength and predicted by the LRF model (M3).

**Figure 4 sensors-21-03373-f004:**
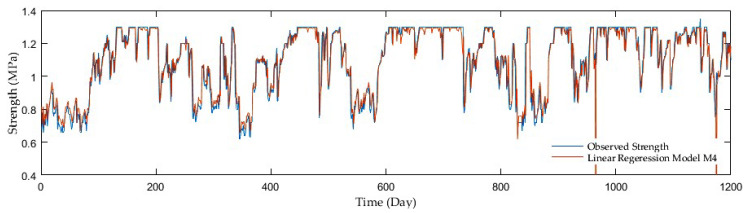
A time series of observed soil strength and predicted by the LRF model (M5).

**Figure 5 sensors-21-03373-f005:**
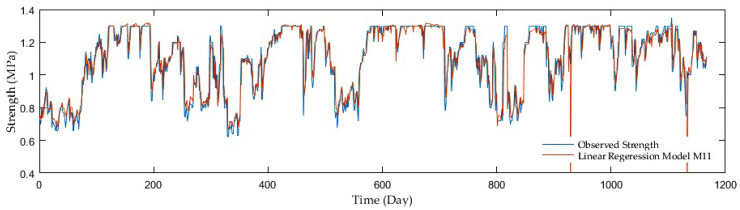
A time series of observed soil strength and predicted by the LRF model (M10).

**Figure 6 sensors-21-03373-f006:**
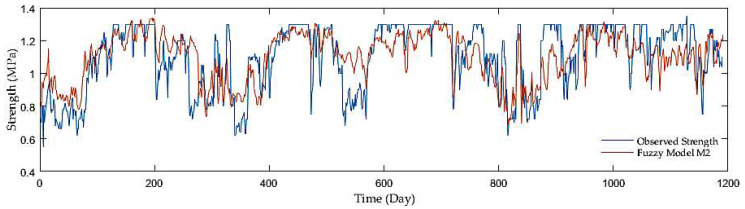
A time series of observed soil strength and predicted by the FL model (M2).

**Figure 7 sensors-21-03373-f007:**
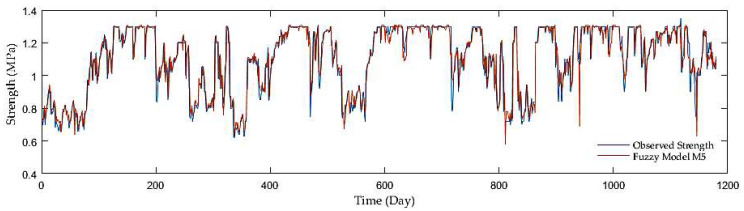
A time series of observed soil strength and predicted by the FL model (M5).

**Figure 8 sensors-21-03373-f008:**
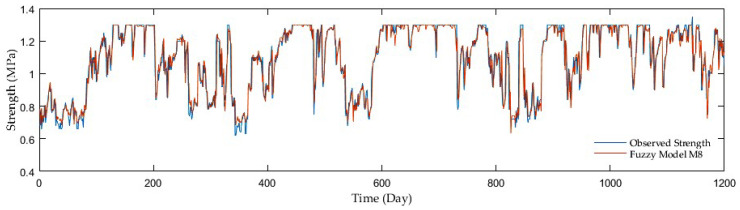
A time series of observed soil strength and predicted by the FL model (M7).

**Figure 9 sensors-21-03373-f009:**
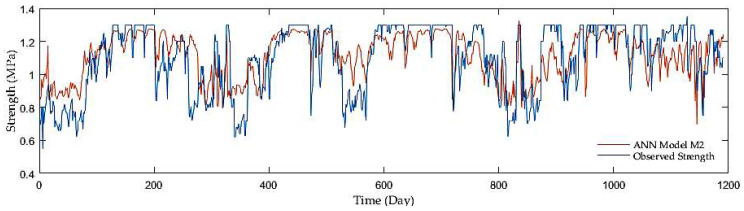
A time series of observed soil strength and predicted by the ANN model (M2).

**Figure 10 sensors-21-03373-f010:**
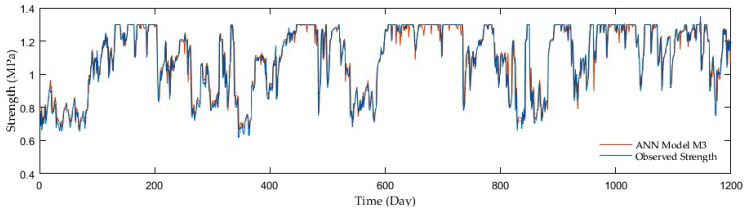
A time series of observed soil strength and predicted by the ANN model (M3).

**Figure 11 sensors-21-03373-f011:**
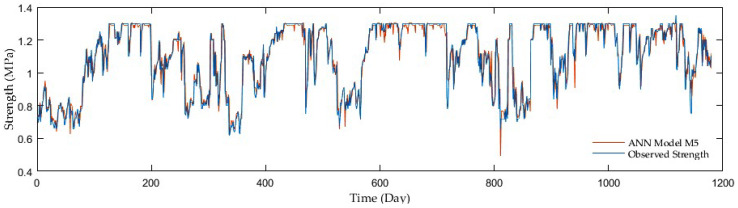
A time series of observed soil strength and predicted by the ANN model (M5).

**Figure 12 sensors-21-03373-f012:**
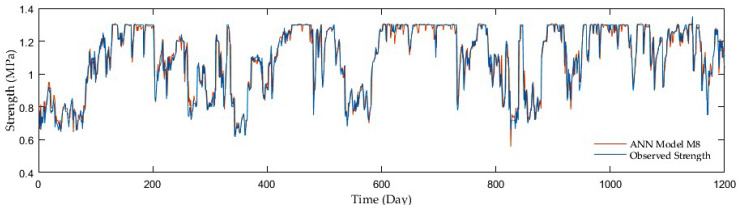
A time series of observed soil strength and predicted by the ANN model (M7).

**Figure 13 sensors-21-03373-f013:**
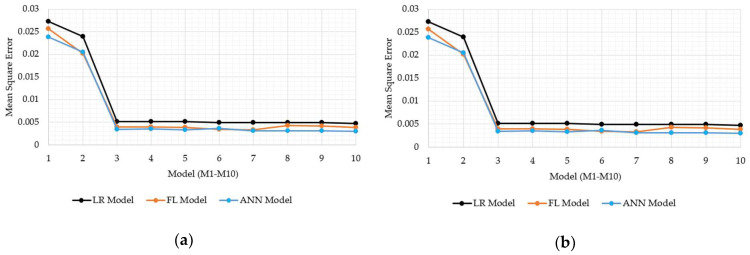
Comparisons of the mean square errors (MSEs) for the LRF, FL, and ANN models; (**a**) calibration data; (**b**) validation data.

**Figure 14 sensors-21-03373-f014:**
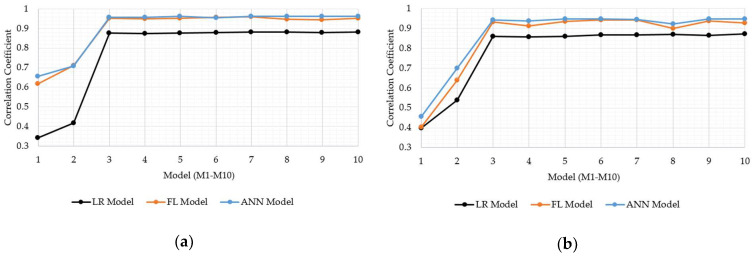
Comparisons of the correlation coefficients (R) for the LRF, FL, and ANN models; (**a**) calibration data; (**b**) validation data.

**Figure 15 sensors-21-03373-f015:**
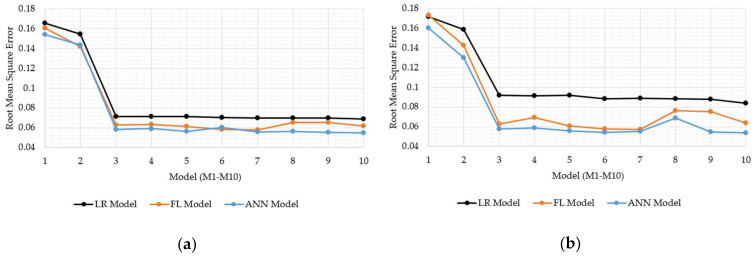
Comparisons of the root mean square errors (RMSEs) for the LRF, FL, and ANN models; (**a**) calibration data; (**b**) validation data.

**Figure 16 sensors-21-03373-f016:**
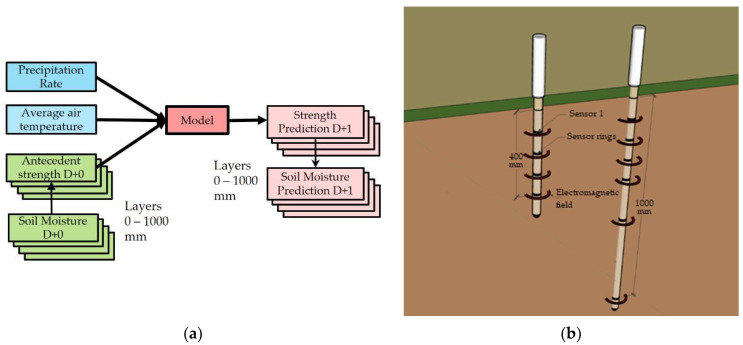
(**a**) A refined model conception; (**b**) soil moisture measurement probes.

**Table 1 sensors-21-03373-t001:** Linear regression model performance evaluation.

Model	Calibration			Validation		
	MSE	R	RMSE	MSE	R	RMSE
Precipitation rate and mean air temperature						
M1	0.027	0.343	0.166	0.03	0.399	0.172
M2	0.024	0.418	0.155	0.025	0.538	0.158
Precipitation rate and antecedent soil strength						
M3	0.005	0.876	0.071	0.008	0.861	0.092
M4	0.005	0.874	0.072	0.008	0.859	0.091
M5	0.005	0.876	0.071	0.008	0.861	0.092
Precipitation rate, mean air temperature, and antecedent soil strength						
M6	0.005	0.88	0.07	0.008	0.868	0.089
M7	0.005	0.881	0.07	0.008	0.868	0.089
M8	0.005	0.881	0.07	0.008	0.869	0.088
M9	0.005	0.879	0.07	0.008	0.866	0.088
M10	0.005	0.881	0.069	0.007	0.873	0.084

**Table 2 sensors-21-03373-t002:** Fuzzy logic model performance evaluation.

Model	Calibration			Validation		
	MSE	R	RMSE	MSE	R	RMSE
Precipitation rate and mean air temperature						
M1	0.026	0.618	0.16	0.03	0.405	0.173
M2	0.02	0.712	0.142	0.02	0.638	0.142
Precipitation rate and antecedent soil strength						
M3	0.004	0.951	0.061	0.004	0.932	0.063
M4	0.004	0.949	0.063	0.005	0.913	0.07
M5	0.004	0.953	0.061	0.004	0.934	0.061
Precipitation rate, mean air temperature and antecedent soil strength						
M6	0.003	0.957	0.058	0.003	0.942	0.058
M7	0.003	0.959	0.058	0.003	0.943	0.057
M8	0.004	0.946	0.065	0.006	0.901	0.077
M9	0.004	0.946	0.065	0.006	0.939	0.075
M10	0.004	0.951	0.062	0.004	0.928	0.064

**Table 3 sensors-21-03373-t003:** Performance evaluation of the artificial neural network models.

Model	Calibration			Validation		
	MSE	R	RMSE	MSE	R	RMSE
Precipitation rate and mean air temperature						
M1	0.024	0.655	0.154	0.026	0.456	0.16
M2	0.021	0.71	0.143	0.017	0.699	0.13
Precipitation rate and antecedent soil strength						
M3	0.003	0.958	0.058	0.003	0.942	0.058
M4	0.004	0.956	0.059	0.004	0.938	0.059
M5	0.003	0.961	0.056	0.003	0.948	0.056
Precipitation rate, mean air temperature and antecedent soil strength						
M6	0.004	0.955	0.06	0.003	0.948	0.054
M7	0.003	0.962	0.056	0.003	0.946	0.055
M8	0.003	0.96	0.056	0.005	0.922	0.069
M9	0.003	0.961	0.055	0.003	0.947	0.055
M10	0.003	0.961	0.055	0.003	0.948	0.054

## Data Availability

The data sets used and analyzed during the current study are available from the corresponding author on reasonable request.

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
