# Peer review of "Applicability of Fuzzy Logic and Artificial Neural Network for Unpaved Airfield Surface Bearing Strength Prediction"

_sensors, 2021, doi:10.3390/s21103373_

Round 1

Reviewer 1 Report

The authors investigated and compared the capabilities of fuzzy logic and artificial neural network with linear regression function for unpaved airfield surface bearing strength prediction. The methodology was described comprehensively and the results are meaningful and valuable. However, there are some problems existing in this paper which the authors must pay attention to deal with.

  1. The significance of this study should be highlighted in the abstract.
  2. In the Introduction, the literature review is not comprehensive. Please study the previous research on this topic with regards to the experimental and theoretical methods in the past 3 years.
  3. Please offer the full name for the term which is mentioned at the first time. For example, the LRF, ANN and FL were mentioned with the full names and their abbreviations for several times in different sections. It is not necessary. The ANN is the abbreviation for “artificial neural network” or “artificial neuron network”?
  4. Some expressions in the manuscript are not clear. An English native speaker is suggested to carefully proofread it again.
  5. The key point of this study is about ANN and FL from the reviewer’s point of view, it is suggested to reduce the content about the introduction and description of the LRF, but increase the content about those of ANN and FL.
  6. Conclusions section is suggested to be added after the discussion section.

Author Response

Comment/suggestion 1: The paper's significance should be highlighted in the abstract.

Author reaction 1: The abstract has been refined with the most relevant and significant points.

Comment/suggestion 2: The literature review is not comprehensive.

Author reaction 2: The introduction has been somewhat revised but the extent remained. I anticipate that the full extent of information is vital to the paper's context despite containing rather old literature references in some cases.

Comment/suggestion 3: The full names and abbreviations of the linear regression function, artificial neural network, and fuzzy logic.

Author reaction 3: The full names of the linear regression function, artificial neural network, and fuzzy logic have been refined. Now they are mentioned only once in the text, and the following text encompasses abbreviations (ANN, LRF, and FL).

Comment/suggestion 4: Proofreading by a native speaker.

Author reaction 4: The text has been corrected by a native speaker.

Comment/suggestion 5: The key point is about ANN and FL.

Author reaction 5: The extent of the text with regards to the linear regression function, artificial neural network has been decreased and altered.

Comment/suggestion 6: Conclusion missing.

Author reaction 6: The conclusion has been added.

Reviewer 2 Report

The paper concerns the applicability of different statistical models for the prediction of the bearing strength of unpaved airfield surfaces trough daily temperature and rainfall data and presents the calibration and validation of these models on a case study. The idea of the research is interesting and presents some novelty. The results could be useful as an aid for airport operation staff to plan and organize the maintenance of unpaved airfield surfaces.

The paper could be improved as follows:

  • in lines 24-27, you talk about the innovative technologies covering aircraft and airport design, from which the need store large amount of information regarding soil bearing capacity to ensure higher and higher safety standards. These topics are often included, especially in airport design and management, in BIM environments, so it would be interesting to enrich by adding more supporting references to the concept of digitalization and integration of pavement data into BIM systems, such as “BIM Approach for Modeling Airports Terminal Expansion. Infrastructures,2020, 5(5), 41” and “BIM Approach for Smart Infrastructure Design and Maintenance Operations. In Transportation Systems for Smart, Sustainable, Inclusive and Secure Cities.2020. IntechOpen”
  • only a few lines are dedicated to the stratigraphy of the soil under analysis (146-149), but it is very important since it defines the range of applicability of your models; you should add some more specifications about the depth of each layer (e.g. silty sands, eroded gneiss etc.) and maybe add a Figure also.
  • lines 184-185: isn’t the prediction also achieved by daily mean air temperature?
  • Equations (9) and (10) present some errors and should be revised
  • the antecedent soil strength has been defined generically as the strength of the soil during the previous measurement. But does the time interval between two consecutive measures affect the quality of the subsequent models? Do the models perform well with daily measurements? In that case you should consider the possilibility of enriching the models with an additional variabile that takes into account that time interval between the measures.
  • you never mention if the statistical analyses have been performed through specific sofware or by a custom script written in a programming environment. Please clarify that.
  • lines 217-218: the variables xi, bk and Ei are not represented in equation (11), please revise.
  • lines 274-279: it doesn’t seem the appropriate section to summarize the explained methodoogy. I suggest you to move these lines at the beginning of Section 2.6
  • Tables 2, 3 and 4 are not easy to read. I suggest you to refer to the model equation number instead of a new number that doesn’t match the previous listed models.
  • line 437: the sentence should not be split. Please revise.
  • Figure 16b indicates two depths but doesn’t show any unit of measurement. Please add (and maybe uniform to the same unit of measurement that you report in Figure 16a)

Author Response

Comment/suggestion 1: Lines 24-27 BIM approach.

Author reaction 1: BIM Approach for Modeling Airports Terminal Expansion has been incorporated into the text.

Comment/suggestion 2: Lines 146 -149 stratigraphy.

Author reaction 2: some specifications about the soil depths were added in paragraph 2.1.

Comment/suggestion 3: Lines 184-185 request to add daily mean air temperature.

Author reaction 3: daily mean air temperature was added into the text.

Comment/suggestion 4: Error in equations (9) and (10).

Author reaction 4: Equations (9) and (10) revised and corrected.

Comment/suggestion 5: the antecedent soil strength has been defined generically as the strength of the soil during the previous measurement. But does the time interval between two consecutive measures affect the quality of the subsequent models? Do the models perform well with daily measurements?

Author reaction 5: A brilliant idea I have not anticipated, I am going to consider the additional variable that covers the time interval between the measures.

Comment/suggestion 6: statistical analyses technique.

Author reaction 6: Information with regards to the statistical analyses whether provided by software or programming script was added and clarified in chapter 2.6. Statistical analyses.

Comment/suggestion 7: Lines 217-218 Missing variables.

Author reaction 7: Variables that were not present were clarified.

Comment/suggestion 8: lines 274-279: it doesn’t seem the appropriate section to summarize the explained methodology. I suggest you to move these lines at the beginning of Section 2.6

Author reaction 8: The text from lines 274-279 removed and the text revised.

Comment/suggestion 9: Tables 2, 3, and 4 are not easy to read.

Author reaction 9: Tables 2, 3, and 4 were altered with proper model names matching with the text.

Comment/suggestion 10: Line 437 the sentence should not be split.

Author reaction 10: The grammatical error was corrected.

Comment/suggestion 11: Figure 16a and b. units are not present.

Author reaction 11: The units were applied and unified for both pictures.